# Comparative effectiveness over time of the mRNA-1273 (Moderna) vaccine and the BNT162b2 (Pfizer-BioNTech) vaccine

Nazmul Islam[1], Natalie E. Sheils[1], Megan S. Jarvis[1] & Kenneth Cohen [1,2✉]

Real-world analysis of the incidence of SARS-CoV-2 infection post vaccination is important in determining the comparative effectiveness of the available vaccines. In this retrospective cohort study using deidentified administrative claims for Medicare Advantage and commercially insured individuals in a research database we examine over 3.5 million fully vaccinated individuals, including 8,848 individuals with SARS-CoV-2 infection, with a follow-up period between 14 and 151 days after their second dose. Our primary outcome was the rate of Covid-19 infection occurring at 30, 60, and 90 days at least 14 days after the second dose of either the mRNA-1273 vaccine or the BNT162b2 vaccine. Sub-analyses included the incidence of hospitalization, ICU admission, and death/hospice transfer. Separate analysis was conducted for individuals above and below age 65 and those without a prior diagnosis of Covid-19. We show that immunization with mRNA-1273, compared to BNT162b2, provides slightly more protection against SARS-CoV-2 infection that reaches statistical significance at 90 days with a number needed to vaccinate of ≥290. There are no differences in vaccine effectiveness for protection against hospitalization, ICU admission, or death/hospice transfer (aOR 1.23, 95% CI (0.67, 2.25)).

[1] Optum Labs, Minnetonka, MN, USA. [2] Optum Center for Research and Innovation, Minnetonka, MN, USA. ✉email: ken.cohen@optum.com

**B**oth the emergence of variants of concern (VOC) and reports of infections post vaccination emphasize the need to study the relative real-world effectiveness of the available vaccines against Covid-19. The mRNA-1273 (Moderna) vaccine and the BNT162b2 (Pfizer–BioNTech) vaccine have both proven highly effective in preventing severe disease, hospitalization, and death from Covid-19. However, data suggest that the humoral antibody response to the vaccines differs, with two doses of the mRNA-1273 vaccine providing significantly higher humoral antibody response compared with two doses of the BNT162b2 vaccine in both uninfected and previously infected individuals across all age categories[1]. This is clinically relevant as several studies have demonstrated that higher humoral antibody responses correlate with enhanced protection against Covid-19[2].

Data also suggest that infection rates follow the same trend as the humoral antibody response. Observational data incorporating cases of Delta VOC have suggested a higher rate of infection in individuals immunized with two doses of the BNT162b2 vaccine compared with two doses of the mRNA-1273 vaccine, suggesting that the real-world performance of the vaccines may differ from one another[3].

We therefore conducted a retrospective cohort study that examined the incidence and severity of Covid-19 infections as a function of the time from vaccination in over 3.5 million individuals across the United States who were fully vaccinated via either the mRNA-1273 or the BNT162b2 vaccine, including over 8800 infections; see Fig. 1. Understanding the clinical performance of each vaccine is critically important to determine the comparative effectiveness of the vaccines.

Our primary outcome is the rate of Covid-19 infection occurring at 30, 60, and 90 days at least 14 days after the second dose of either the mRNA-1273 vaccine or the BNT162b2 vaccine.

## Results

In a population of over 3.5 million fully vaccinated individuals, 8848 experienced documented Covid-19 infections. Of those, 3090 (35%) received mRNA-1273 and 5758 (65%) received BNT162b2 (Table 1 and Fig. 1).

Median age and prevalence of many comorbid conditions were similar across groups considered (Table 1). Models for the seven negative outcomes, included to assure there were no unaccounted-for differences between groups receiving BNT162b and mRNA-1273, exhibited negligible mean bias arising from systemic errors (Table S3).

Kaplan–Meier curves report that the estimated time to infection and time to severe negative outcomes defined as any hospitalization, ICU admission, death, or discharge to hospice, whichever occurs first, are reported in Fig. 2. In the raw data, a statistically significant difference ($p$-value < 0.001) with a diverging trend over time is observed between BNT162b and mRNA-1273 (Fig. 2C). This

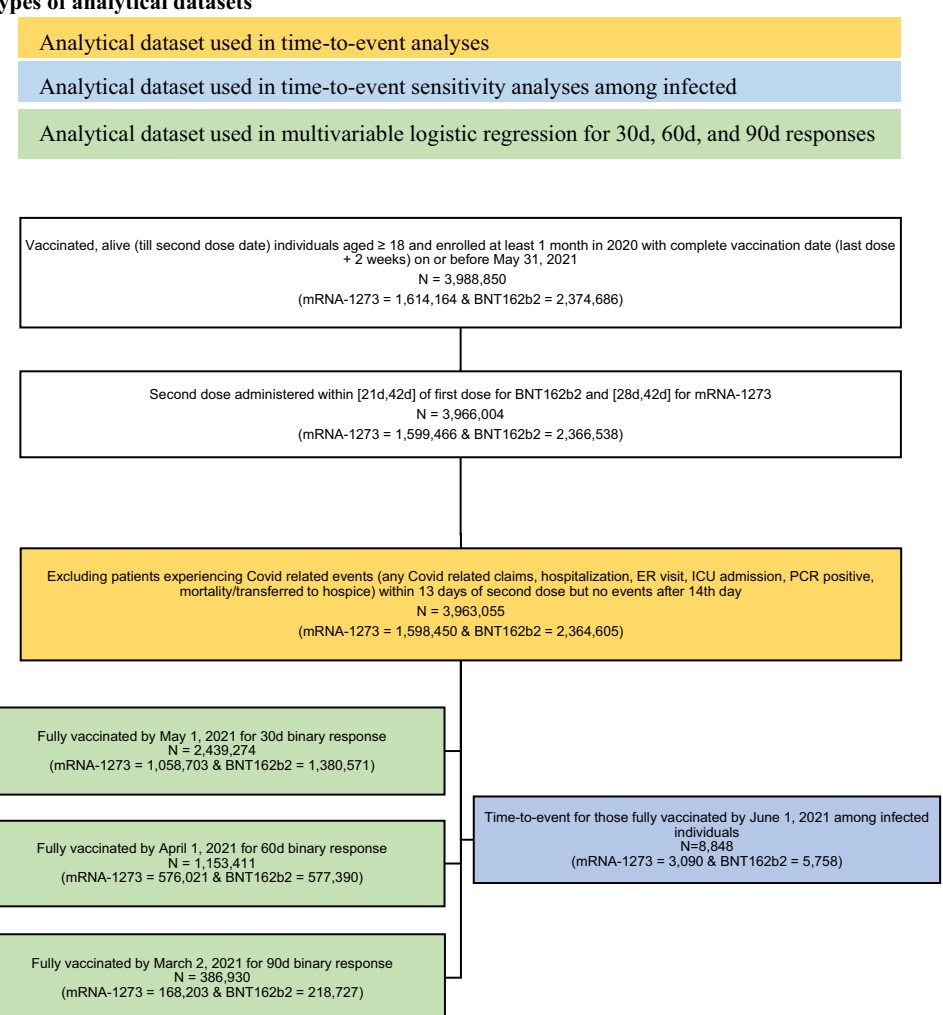

**Types of analytical datasets**

Analytical dataset used in time-to-event analyses

Analytical dataset used in time-to-event sensitivity analyses among infected

Analytical dataset used in multivariable logistic regression for 30d, 60d, and 90d responses

Vaccinated, alive (till second dose date) individuals aged ≥ 18 and enrolled at least 1 month in 2020 with complete vaccination date (last dose + 2 weeks) on or before May 31, 2021
N = 3,988,850
(mRNA-1273 = 1,614,164 & BNT162b2 = 2,374,686)

Second dose administered within [21d,42d] of first dose for BNT162b2 and [28d,42d] for mRNA-1273
N = 3,966,004
(mRNA-1273 = 1,599,466 & BNT162b2 = 2,366,538)

Excluding patients experiencing Covid related events (any Covid related claims, hospitalization, ER visit, ICU admission, PCR positive, mortality/transferred to hospice) within 13 days of second dose but no events after 14th day
N = 3,963,055
(mRNA-1273 = 1,598,450 & BNT162b2 = 2,364,605)

Fully vaccinated by May 1, 2021 for 30d binary response
N = 2,439,274
(mRNA-1273 = 1,058,703 & BNT162b2 = 1,380,571)

Fully vaccinated by April 1, 2021 for 60d binary response
N = 1,153,411
(mRNA-1273 = 576,021 & BNT162b2 = 577,390)

Fully vaccinated by March 2, 2021 for 90d binary response
N = 386,930
(mRNA-1273 = 168,203 & BNT162b2 = 218,727)

Time-to-event for those fully vaccinated by June 1, 2021 among infected individuals
N=8,848
(mRNA-1273 = 3,090 & BNT162b2 = 5,758)

**Fig. 1 Description of the analytical datasets with study-selection criteria and counts by each vaccine.** Legends (orange, blue, and green color shaded) are provided to highlight the analytical datasets with differing sample size.

**Table 1 Descriptive statistics for fully vaccinated individuals.**

| | 30 DAY POST-VACCINATION BINARY OUTCOME | | 60 DAY POST-VACCINATION BINARY OUTCOME | | 90-DAY POST-VACCINATION BINARY OUTCOME | | POST-VACCINATION TIME-TO-EVENT | |
|---|---|---|---|---|---|---|---|---|
| | BNT162b2 | mRNA-1273 | BNT162b2 | mRNA-1273 | BNT162b2 | mRNA-1273 | BNT162b2 | mRNA-1273 |
| N | 1,380,571 | 1,058,703 | 577,390 | 576,021 | 218,727 | 168,203 | 2,364,605 | 1,598,450 |
| Age, mean (SD) | 56.17 (17.30) | 59.82 (17.42) | 61.14 (18.25) | 62.62 (17.67) | 57.16 (19.30) | 56.14 (18.34) | 50.67 (17.46) | 55.39 (17.84) |
| Elixhauser readmission score, mean (SD) | 5.95 (11.64) | 6.87 (12.22) | 7.76 (13.66) | 7.63 (12.86) | 7.68 (14.59) | 6.05 (11.76) | 4.65 (10.16) | 5.83 (11.32) |
| Transferred from nursing facility/ SNF (%) | 2,368 (0.2) | 719 (0.1) | 2,236 (0.4) | 609 (0.1) | 1873 (0.9) | 420 (0.2) | 2465 (0.1) | 832 (0.1) |
| Sex | | | | | | | | |
| Female (%) | 780,162 (56.5) | 608,628 (57.5) | 354,134 (61.3) | 344,902 (59.9) | 143,348 (65.5) | 105,587 (62.8) | 1,264,075 (53.5) | 874,407 (54.7) |
| Male (%) | 600,409 (43.5) | 450,075 (42.5) | 223,256 (38.7) | 231,119 (40.1) | 75,379 (34.5) | 62,616 (37.2) | 1,100,530 (46.5) | 724,043 (45.3) |
| Residence by region (%) | | | | | | | | |
| Midwest | 190,544 (13.8) | 155,541 (14.7) | 77,097 (13.4) | 80,167 (13.9) | 33,699 (15.4) | 32,208 (19.1) | 335,877 (14.2) | 228,427 (14.3) |
| Northeast | 395,294 (28.6) | 304,757 (28.8) | 172,651 (29.9) | 164,256 (28.5) | 59,359 (27.1) | | 672,676 (28.4) | 467,702 (29.3) |
| South | 439,164 (31.8) | 323,742 (30.6) | 185,910 (32.2) | 172,565 (30.0) | 67,205 (30.7) | 64,696 (38.5) | 752,567 (31.8) | 481,238 (30.1) |
| West | 355,569 (25.8) | 274,663 (25.9) | 141,732 (24.5) | 159,033 (27.6) | 58,464 (26.7) | | 603,485 (25.5) | 421,083 (26.3) |
| First dose administered before Feb 1, 2021* (%) | 317,863 (23.0) | 359,372 (33.9) | 317,863 (55.1) | 359,372 (62.4) | 86,899 (39.7) | 82,298 (48.9) | 317,863 (13.4) | 359,372 (22.5) |
| SES index, mean (SD) | 53.36 (2.85) | 53.12 (2.80) | 52.68 (3.15) | 52.91 (2.80) | 52.81 (3.34) | 53.70 (2.90) | 53.08 (3.12) | |
| Study-related outcomes | | | | | | | | |
| Infection (%) | 3766 (0.3) | 1949 (0.2) | 3376 (0.6) | 1724 (0.3) | 2514 (1.1) | 861 (0.5) | 5758 (0.2) | 3090 (0.2) |
| Hospitalization/ ICU/ deceased/ transferred to hospice (%) | 72 (0.0) | 42 (0.0) | 82 (0.0) | 49 (0.0) | 55 (0.0) | 23 (0.0) | 129 (0.0) | 85 (0.0) |
| ICU/ deceased/ transferred to hospice (%) | 32 (0.0) | 14 (0.0) | 36 (0.0) | 21 (0.0) | 26 (0.0) | 8 (0.0) | 57 (0.0) | 40 (0.0) |
| Hospitalization (%) | 58 (0.0) | 35 (0.0) | 67 (0.0) | 39 (0.0) | 46 (0.0) | 21 (0.0) | 105 (0.0) | 69 (0.0) |
| ICU (%) | 28 (0.0) | 14 (0.0) | 30 (0.0) | 20 (0.0) | 20 (0.0) | 7 (0.0) | 47 (0.0) | 38 (0.0) |
| Deceased/ transferred to hospice (%) | 9 (0.0) | 2 (0.0) | 12 (0.0) | 1 (0.0) | 10 (0.0) | 1 (0.0) | 21 (0.0) | 7 (0.0) |

Descriptive statistics for baseline characteristics (demographics, socioeconomic, comorbidities, negative controls, study outcomes, and other prognostic factors) among fully vaccinated individuals by BNT162b2 and mRNA-1273. Analytical datasets are separated into those fully vaccinated by May 1, 2021, April 1, 2021, and March 2, 2021, ensuring each patient had at least 30, 60, and 90 days of post-vaccination follow-up to experience events, respectively. For time-to-event outcomes we consider individuals vaccinated by June 1, 2021.
Note: A more comprehensive descriptive table (Table S1) is provided in the supplementary section.
*For the 90-day outcome the date was adjusted to "First dose administered before January 10, 2021."

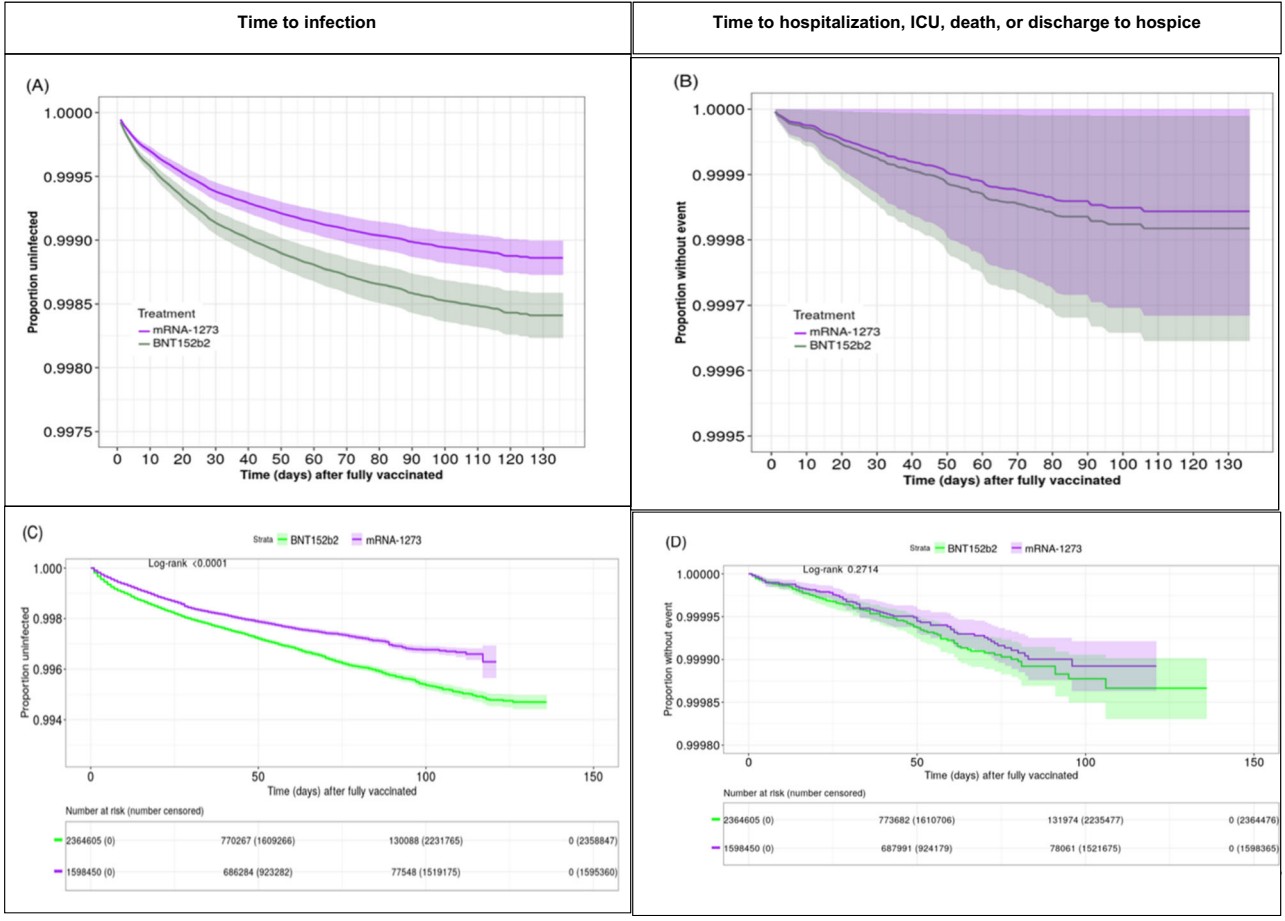

**Fig. 2 Predicted survival probabilities.** Panels (**A**) and (**B**) show predicted survival probabilities based on the univariate model with inverse probability of treatment weight (IPTW) for (A) SARS-CoV-2 infection and (**B**) hospitalization/ICU/death/hospice, whichever occurs first. Raw data are shown in (**C**) and (**D**) via Kaplan–Meier curves for time-to-events among all individuals vaccinated by June 1, 2021, for the event of (**C**) infection, (**D**) hospitalization/ICU/ death/hospice, whichever occurs first.

difference persists in the PH model-based predictions for a general patient (Fig. 2A). There are no statistically significant differences between the predicted probabilities of the two vaccines (Fig. 2B, D) for the composite outcome of hospitalization, ICU admission, death, or transfer to hospice.

Adjusted odds ratios for likelihood of infection 30, 60, and 90 days post vaccination (14 days after the second dose) show that the mRNA-1273 vaccine is associated with lower odds of infection. For the unadjusted model with IPTW as sampling weights, the NNV to observe this difference ranges from 1,047 over 30 days to 290 over 90 days (Fig. 3), signaling superior mRNA-1273 vaccine effects over time in reducing the likelihood of infection. This difference is consistent across all the models considered. However, for severe adverse outcomes (ICU admission, composite ICU admission/death/referral to hospice, or composite hospitalization/ICU admission/death/referral to hospice), while events are rare, no statistically significant differences between the two vaccines were observed (Fig. S4–6). As shown in Fig. 4, time-to-event analysis assuming right censoring, was performed and provided similar results to the binary outcome results for different model specifications, including for outcomes of infection (aHR 0.69, 95% CI (0.66, 0.72)), composite ICU admission/death/referral to hospice (aHR 0.76, 95% CI (0.50, 1.16)), and composite hospitalization/ICU admission/death/referral to hospice (aHR 0.67, 95% CI (0.51, 0.89)). Risk differences between mRNA-1273 and BNT162b2 for both infection and composite outcomes increase over time (Fig. 4b), this observation is in alignment with the binary outcome results (Fig. 3).

We further considered time-to-event analyses among all those experiencing infections in our sample for ICU admission, composite ICU admission/death/referral to hospice, and composite hospitalization/ICU admission/death/referral to hospice. No statistically significant differences between vaccines were observed for adverse composite outcomes (Figs. S12–14).

The results are similar for the stratified analyses of (1) including only patients with no prior diagnosis of Covid-19 (Fig. S11), (2) including only patients who are aged 65 or older (Fig. S9), and (3) including only patients who are aged 64 and younger (Fig. S10). Since the population considered is different in each model, between-model comparisons are not valid, however, both models show directionally the same results.

It is known that certain comorbidities worsen prognosis in an unvaccinated population. Our models indicate that in a vaccinated population, congestive heart failure (aHR 1.52, 95% CI (1.03, 2.26)), hypertension (aHR 2.17, 95% CI (1.30, 3.62)), and lymphoma (aHR 7.03, 95% CI (4.31,11.47)) increase the likelihood of experiencing the composite adverse event of hospitalization, ICU admission, or death/ transfer to hospice. This difference persisted even after adjusting for which vaccine a person received, sociodemographic variables, transfer from nursing facility, timing of vaccination, place of residence, and historical comorbidities (Fig. S7).

## Discussion

Multiple studies have shown that the incidence of SARS-CoV-2 infection in vaccinated individuals has increased over the last

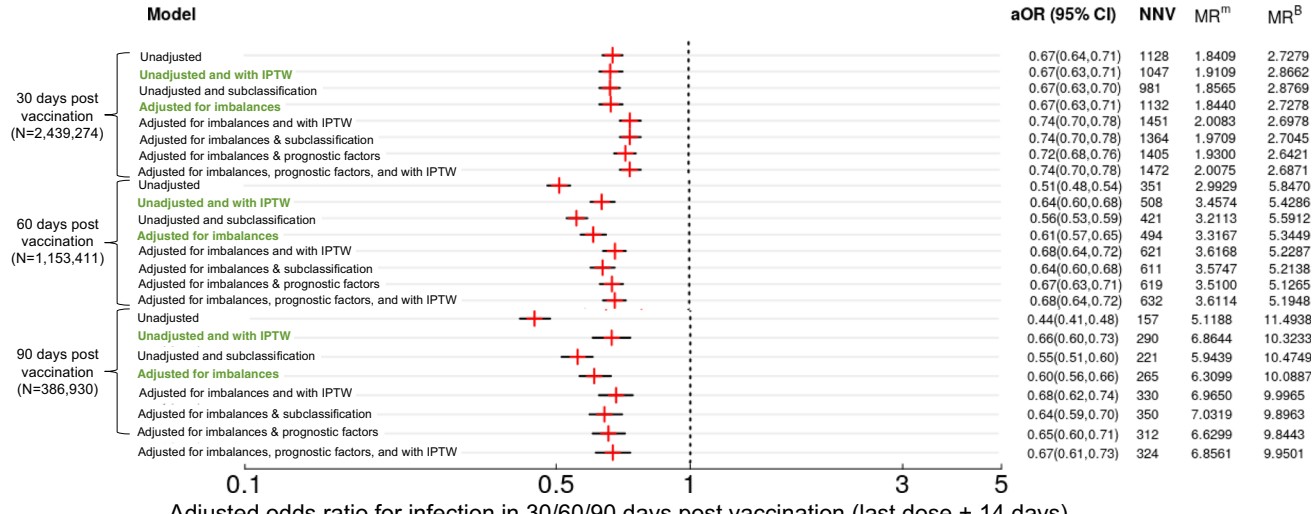

**Fig. 3 Outcomes of vaccination.** Adjusted odds ratio along with 95% confidence intervals, number needed to vaccinate (NNV), and marginal rates (MR) of infection per 1000 people within 30/60/90 days post vaccination (last dose + 14 days) for BNT162b2 (denoted with a superscript of B) and mRNA-1273 (denoted with a superscript of m). Note that eight models are considered for each outcome, "30 days post vaccination", "60 days post vaccination" and "90 days post vaccination". (1) Unadjusted: a univariate model adjusting only for vaccine type. (2) Unadjusted and with IPTW: weighted univariate model adjusting only for vaccine type. (3) Unadjusted and subclassification: univariate model, stratified by quintiles of propensity scores, adjusting only for vaccine type. (4) Adjusted for imbalances: multivariable model adjusting for age, timing of vaccine, residence, prior history of Covid-19 diagnosis, and socioeconomic status. (5) Adjusted for imbalances with IPTW: weighted multivariable model adjusting for age, timing of vaccine, residence, prior history of Covid-19 diagnosis, and socioeconomic status. (6) Adjusted for imbalances and subclassification: multivariable model, stratified by quintiles of propensity scores, adjusting for age, timing of vaccine, residence, prior history of Covid-19 diagnosis, and socioeconomic status. (7) Adjusted for imbalances, prognostic factors: multivariable model adjusting for all the above variables plus nursing facility residence and comorbidities that are selected in the variable-screening step. (8) Adjusted for imbalances, prognostic factors with IPTW: weighted multivariable model adjusting for all the above variables plus nursing facility residence and comorbidities that are selected in the variable-screening step.

year. This increase may be related to the higher transmissibility of the Delta and Omicron VOC and/or the potential contribution of waning immunity post vaccination. These data underscore the need to understand the comparative effectiveness of the available vaccines.

Two important questions must be answered to address the comparative effectiveness of the BNT162b and mRNA-1273 vaccines. The first is whether each vaccine is equivalent in its ability to prevent severe disease from Covid-19. The data available from our large population of patients suggest that at 90 days out from vaccination, there is no significant difference between BNT162b and mRNA-1273 in terms of the risk of the composite outcome of hospitalization, ICU admission, or death/transfer to hospice (aOR [95% confidence interval]: 1.23 [0.67, 2.25], Fig. S6). We analyzed data both from a binary and time-to-event outcome point of view, each quantifying the association between patient outcomes and vaccine types with different model specifications. Conclusions remain similar, irrespective of modeling framework, highlighting the robustness of our methods. The second question is whether there is a higher incidence of SARS-CoV-2 infections with the BNT162b vaccine compared with the mRNA-1273 vaccine, and here there is a statistically significant difference in favor of the mRNA-1273 vaccine. This difference appears very early post vaccination and increases over time. The NNV with the mRNA-1273 vaccine compared with the BNT162b vaccine to prevent one case of SARS-CoV-2 is 1047 at thirty days post vaccination (aOR [95% CI]: 0.67 [0.63, 0.71]), but is 290 at ninety days post vaccination (aOR [95% CI]: 0.66 [0.60, 0.73]). Although this incremental risk is small at the individual level, it is meaningful at the population level. Our results suggest that for every 1 million individuals vaccinated with the BNT162b vaccine compared with the mRNA-1273 vaccine, this would represent 3,448 additional care-seeking cases of Covid-19 at 90 days.

Delta infection was of very low prevalence when these data were collected, and therefore unlikely affected our comparative effectiveness results. However, as new VOCs emerge, each of these may have unique characteristics that would impact their ability to evade vaccine-induced immunity. Periodic updates to this analysis will allow us to track the comparative effectiveness of mRNA-1273 and BNT162B in the setting of the emergence of new VOCs, including Delta and Omicron.

This study has limitations. First, the analysis is restricted to commercially insured and Medicare Advantage beneficiaries from a single U.S. insurer, a group that is unevenly distributed across the United States geographically and demographically. Nevertheless, this study reflects a large and comprehensive sample of U.S. vaccinations. Second, we are unable to measure SARS-CoV-2 infection that is not apparent in medical claims or via laboratory testing, which likely results in an overestimate of the vaccines' protective effects. This includes the fact that our "time-to-infection" represents date of infection from our data (positive PCR test, or ICD-10 code of U07.1 in claims), rather than the date on which SARS-CoV-2 was contracted. Notably, asymptomatic or mild disease for which an individual did not seek care are not captured. However, serious adverse events, which pose the most strain on the healthcare system and for individuals, are reliably observable in our data and are less affected by diagnosis-dependent biases over time. Finally, we do not include an analysis accounting for calendar time as a proxy to infer the likely variant of infection as this additional confounder would further complicate already-nuanced results. While not undertaken here, further study to understand the relative effectiveness against all variants of concern is needed and will be conducted when our data on infection with Delta and Omicron are more complete. Basic counts are included in Table S8.

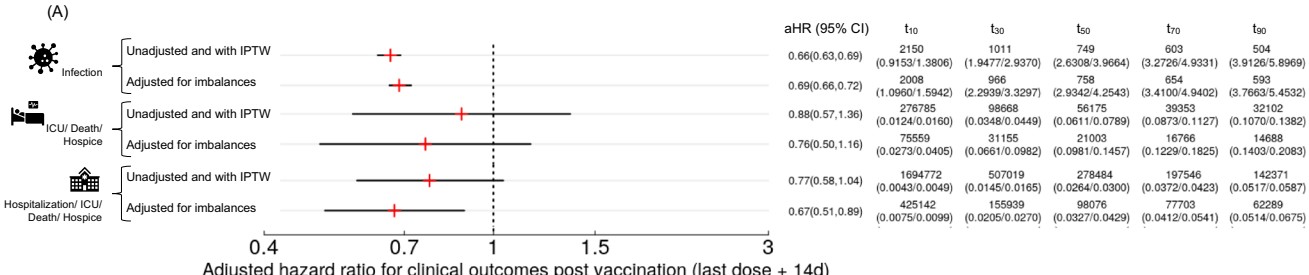

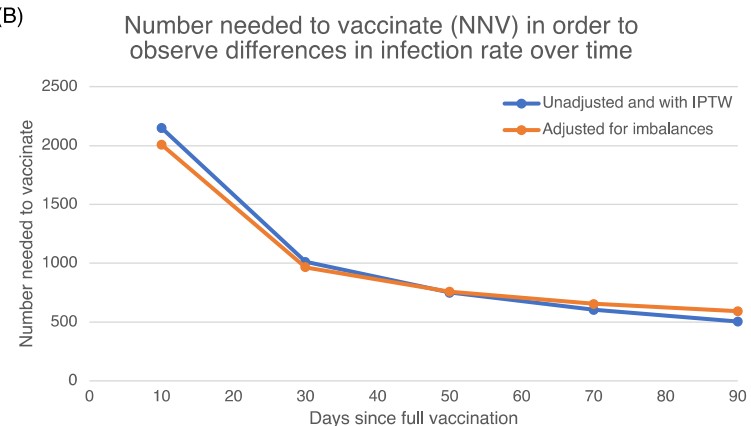

**Fig. 4 Expanded outcomes of vaccination. A** Adjusted hazard ratios (aHRs) along with 95% confidence intervals, number needed to vaccinate (NNV), and predicted marginal rates per 1,000 people (in parenthesis) of experiencing events at 10d, 30d, 50d, 70d, and 90d post vaccination (last dose + 14d) for mRNA-1273 and BNT162b2, respectively. Time-to-event for (top-panel) infection, (middle-panel) serious adverse outcome of ICU admission/ hospitalization/death/transfer-to-hospice, and (bottom-panel) adverse outcome of hospitalization/ICU admission/death/transfer-to-hospice whichever occurs first among all vaccinated individuals. **B** Number needed to vaccinate to observe a difference in infection rates between mRNA-1273 and BNT162b2 at 10d, 30d, 50d, 70d, and 90d post vaccination (last dose + 14d). Note that two models are considered for each outcome. (1) Unadjusted and with IPTW: weighted univariate model adjusting only for vaccine type. (2) Adjusted for imbalances: multivariable model adjusting for age, timing of vaccine, residence, prior history of Covid-19 diagnosis, and socioeconomic status.

This study also has strengths. It represents a geographically and sociodemographically diverse group of 3,966,004 patients, allowing confidence in the estimation of individual-level patient factors associated with documented breakthrough SARS-CoV-2 infection and the resulting serious adverse events.

The incidence of SARS-CoV-2 infection in this large cohort of individuals post full vaccination with mRNA-1273 or BNT162b suggests that the effectiveness of the mRNA-1273 vaccine exceeds that of the BNT162b vaccine by a small margin. However, both vaccines compared equally with respect to the incidence of severe disease at 90 days defined by hospitalization, ICU admission, discharge to hospice, or death (Fig. S6; aOR 1.23, 95% CI (0.67, 2.25)).

## Methods

This paper follows STROBE reporting guidelines for cohort studies. This study was deemed exempt from institutional board review by the UnitedHealth Group Office of Human Research Affairs due to using deidentified retrospective data.

**Data sources**. We used administrative deidentified claims for Medicare Advantage and commercially insured individuals in a research database, including vaccination status through May 31, 2021. This study was reviewed and deemed exempt by the institutional review board of UnitedHealth Group.

**Population**. Starting with all Medicare Advantage or commercial enrollees 18 years or older vaccinated after emergency-use authorization (EUA) for either mRNA-1273 or BNT162b2 who were fully vaccinated (received their second dose with 14 days of additional observation) on or before May 31, 2021. Those who received a second dose earlier than EUA-approved timeframe (21 days for BNT162b2 and 28 days for mRNA-1273) or more than 42 days after their first dose were excluded from

analyses. We then required valid zip-code information and excluded any patients who experienced a Covid-related event (ICD-10 code U07.1 in inpatient or outpatient setting or positive PCR laboratory test for SARS-CoV-2) within 13 days of their second dose and none after "full vaccination," which is considered 14 days after the second dose, to eliminate those whose infections occurred before they were fully vaccinated (Fig. 1). For subsequent analyses, we subdivided this group into those who had at least 30, 60, and 90 days of follow-up observation. In the supplementary information, we also look in particular at the subset of 8848 fully vaccinated individuals who became infected with Covid-19 (identified either by a positive PCR test or ICD-10 code U07.1, including ER, outpatient, and inpatient hospital visits).

**Outcome measure**. Our main outcome was SARS-CoV-2 infection identified either by a positive PCR test or Covid-19 ICD-10 code U07.1 in claims, including ER, outpatient, and inpatient hospital visits. We define this as a "care-seeking" population. We also considered ICU admission, hospital admission, and the composite of either inpatient mortality or discharge to hospice within 30, 60, and 90 days of initial admission for Covid-19. We considered this composite measure a more complete representation of the outcome of interest than mortality alone as it reflects an outcome closer to any-site mortality, and given known racial differences in hospice use[4].

**Statistical analysis**. Binary outcomes such as SARS-CoV-2 infection, hospital admission, composite outcomes such as ICU, death, or hospice transfer with and without hospitalization that occurred within 30-, 60-, and 90 days post vaccination were considered. A series of multivariable logistic regressions[4] were performed to estimate the odds of experiencing events for those vaccinated with mRNA-1273 compared with BNT162b2 adjusting for risk factors (demographic, nursing facility admission source, socioeconomic status (SES) index, comorbidities, time of vaccination, residence by state, prior Covid diagnosis, and urban/rural location). To mitigate any potential selection bias, sample propensity scores, signaling conditional probability of receiving mRNA-1273 and BNT162b2, were estimated via multivariable generalized linear model with logit link function and inverse probability of treatment weights (IPTWs) were calculated using stabilization[5,6]. Both weighted univariate and multivariable logistic regressions[7–12] were performed to validate the

results of unweighted samples. As a sensitivity, propensity-score subclassification models were fitted with respect to quintiles of the estimated propensity scores[13,14].

Kaplan–Meier analysis was performed for time-to-infection and time-to-composite outcomes for individuals fully vaccinated by June 1, 2021; log-rank-based p-values were provided. Multivariable Cox proportional hazard (PH) models[15] were fitted with and without IPTWs. We performed three-sensitivity analyses by (a) stratifying data for vaccinated individuals aged 65 and above, (b) stratifying data for vaccinated individuals under age 65, and (c) excluding patients with a history of any prior Covid-19 diagnoses before complete vaccination. PH model-based predictions of experiencing events for a mRNA-1273- and a BNT162b2-vaccinated individual were provided.

Weighted standardized mean differences and statistical significance tests based on weighted regressions[16] were used to check balances within measured covariates. Standard errors of the parameters were estimated by robust sandwich covariance estimators for all models. Mean predicted marginal probabilities (risk) were calculated via "recycled" predictions[17–19], assuming everyone in the sample received each vaccine, then multiplied by 1000 to convert to predicted marginal risk per 1000 individuals, and number needed to vaccinate (NNV) was obtained with respect to absolute risk differences based on each model[19,20].

As sensitivity, we reported "E-values"[21] to assess the magnitude of unmeasured confounding. Summary measures[22] to determine the strength of selection bias were provided for main outcomes of interest. Multivariable generalized linear models with respect to seven negative-control outcomes were fitted and empirical null distribution of systematic error was used to quantify the mean of bias[23,24]. All statistical tests were two-sided, with a significance level of 5%. All analyses were conducted using R, version 3.6.3[25]. Technical details are included in the Supplementary Information.

**Reporting summary**. Further information on research design is available in the Nature Research Reporting Summary linked to this article.

## Data availability

The raw data are protected and are not available due to data-privacy laws. The processed data are available at reasonable request to the corresponding author.

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

## Acknowledgements

The authors wish to thank Alyssa Wong for her contributions to helping us identify negative outcomes.

## Author contributions

N.E.S., N.I., and K.C. designed the study and drafted the paper. M.S.J. processed and pulled the data. N.I. performed the data analysis. All authors interpreted the results of the analysis, reviewed and revised the paper, and agreed to its publication.

## Competing interests

The authors declare no competing interests.
