## [Peer Review File · Nature Communications]

Comparative Efficacy over time of the mRNA-1273 (Moderna) vaccine and the BNT162b2 (Pfizer-BioNTech) vaccineReviewers' Comments:

Reviewer #1:

Remarks to the Author:

Summary: This paper compared effectiveness of Moderna and Pfizer vaccines vs. infections detected at health care centers and composite serious/fatal disease endpoint using USA data from Medicare advantage and commercial insurance providers. This covered >3.5m vaccinated individuals, of which 6434 had SARS-COV-2 infections between 14 and 151 days after 2nd dose. They used Cox regression time to event models, considering covariate adjustment and inverse probability weighting (IPW) to adjust for potential confounders, and performed secondary analyses on >65yr and on not previously infected cohorts. The infection rate for Moderna was lower than Pfizer, initially 20-25% lower but increasing after 90 days to nearly 50% lower. Similar effect sizes observed for composite serious disease endpoint, but not statistically significant.

General Comments:

This paper presents useful relative effectiveness results for the two key mRNA vaccines used in the USA, Moderna and Pfizer, with respect to infection in those with health-seeking behavior and a composite serious disease index. The large data set with extensively documented demographic information and comorbidities along with vaccination date enable this rigorous analysis, that uses advanced causal inference tools to adjust for potential confounding.

The results have potential import in providing relative effectiveness information, although the ability of the results to inform prioritization and decision making for boosters is limited by the fact that results did not compare with unvaccinated controls or compare early/late periods to assess decrease magnitude of potential immune protection. Further, the study does not adjust for infection time effects, especially as related to the emergence of Delta in the summer. This could potentially confound the inter-vaccine comparison, and clearly Delta is confounded with time since vaccination, which makes the results more difficult to interpret. This should at a minimum be discussed more thoroughly, and preferably adjusted for in the analysis. Also, a sub analysis including days 0-13 and another for previously infected subset, although underpowered, would provide useful point estimates that would be relevant and useful.

Specific comments:

- The details of the IPTW model should be presented in the supplement.
- What are the characteristics of the Medicare Advantage and Commercially Insured cohorts? They are combined together, but are fundamentally different in many ways. The authors should:
 - o Put together a table of demographic and clinical factors for the two cohorts to show their differences.
 - o Perform a secondary analyses on each of these cohorts separately to see whether the vaccine differences, and the vaccine by time interaction, differ across these cohorts.
- Minor point: Table 1 should split out age by 5-year groups, not just mean and variance, to show the distribution of this key factor across vaccine groups. Do the same for the supplemental tables split by Medicare/commercial cohorts.
- Show p-values for comparisons in Table 1 between the vaccinated group for various factors (could be a supplemental table if it doesn't fit in the main table)
- The presented results focus on the NNV, which is useful but whose scale depends heavily on the context and is difficult to compare with results from other studies looking at relative rates. The text and abstract should also present the HR and 95% CI for each comparison. For the non-statistically-significant composite index, still present the estimated HR and 95% CI to show the estimated effect size and precision. I know that the supplement contains results for many different model choices/adjustments – pick whichever one is considered the primary one and make that the one whose HR and 95% intervals you present.
- It is helpful that the secondary analysis of “not previously infected” was done, but the authors should also add a secondary analysis including only the “previously infected” cohort. This will be smaller sample size and underpowered but given the lack of data on this endpoint in many existing

studies, even a point estimate and 95% CI could help demonstrate vaccine differences in previously infected.

- Similarly for the age stratification – a separate secondary analysis was done in the >65 in addition to the overall. A <=65 cohort analysis should also be added to the secondary analysis section in the supplementary materials.
- The modeling started at 14 days after 2nd dose, which is standard since that is when the “full protection” kicks in. There is a paucity of data on advanced endpoints like hospitalization, ICU, death in your composite endpoint from day 0 of dose 1 until day 14 to dose 2, and certain people in society and social media try to claim this is done to conceal evidence of dangerous vaccine side effects. Please also include a secondary analysis of your infection and especially composite serious disease endpoint during this time frame – even if underpowered, the point estimates and transparency of showing it will be a contribution to the literature, in my opinion.
- The model rigorously adjusts for many of the key confounders, which is a significant strength of the paper. However, there is no adjustment for time of infection, a key confounder especially given the emergence of Delta in summer 2021. Is there some way to adjust for time confounding in the modeling?
 - Related to this concern about time effects, I have the following questions/suggestions/requests:
 - o What is the time frame of follow up period, and in particular does it include Delta periods and pre-Delta periods?
 - o Are potential calendar time effects modeled? It doesn't look like it.
 - o Is the distribution of vaccination times over time for the two vaccines consistent? Plot a histogram of vaccinations over time for the two to see if they match up over time. If very different, this is a potential confounder and should be discussed.
 - While the comparison between the two vaccines is useful, and seeing the difference increases over time is especially interesting, the lack of formal comparison across time regions since vaccination and the lack of unvaccinated controls makes it more difficult to interpret.
 - o Ideally, it would be great to pull out a matched unvaccinated cohort against which to compare each of the vaccines and time since vaccination to give an idea of vaccine effectiveness and its waning over time. It is not clear whether after waning the Pfizer loses most all of its effectiveness or not. I recognize this might be a lot of work, but would add substantially to the paper if done.
 - o The days 30-60 and 60-90 infection rates should be compared with the days 0-30 after 14 days post vaccination infection rates to quantify the increased hazard ratio evident in the waning/Delta effect.
 - By the way, the comparison of days 60-90 to days 0-30 could be due to waning, Delta, or some combination. This should be thoroughly discussed, and if possible, proportion of delta in the population used as a covariate in the analysis.
 - Discussion line 249 “potential risks for booster doses, particularly if immune related side effects more frequent than prevented serious infections” – transient side effects not equivalent to hospitalized side effects – reword and if suggesting vaccines increase probability of serious adverse events, provide citation – the clinical trials did not find this relative to placebo in either study.

Reviewer #2:

Remarks to the Author:

This paper compares differential vaccine efficacy of between the mRNA-1273 (Moderna) and the BNT162b (Pfizer) COVID-19 vaccines. The analysis is based on 3.5 million patient records from a single US insurer. The analyses controlled for many factors using appropriate statistical methods (e.g., using multivariable logistic regression with inverse probability of treatment weights based on propensity scores). The analyses were very thorough, and included sensitivity analyses to explore the assumptions. The paper addresses an important issue, and the analyses help shed some light on it. I have a few comments.

Major Comments

1. I feel the study limitations paragraph in the discussion section could be expanded. My main concern is that there may have been some important differences between the vaccine groups that were not measured (i.e., there may have been some unmeasured confounders). One possible unmeasured confounder is degree of urban/rural of residence. One major difference between the vaccines is the necessary refrigeration. The BNT162b vaccine requires more intense refrigeration, and some rural hospitals could not afford that (see e.g., <https://www.statnews.com/2020/11/11/rural-hospitals-cant-afford-freezers-to-store-pfizer-covid19-vaccine/>). So there may be an urban/rural difference in the distribution of the two vaccines. Although the authors mentioned adjusting for residence by state, within each state there are both urban and rural areas. Additionally, there may have been a substantial urban/rural difference in SARS-CoV-2 infection rates during the time the study was covering. Perhaps you could discuss this possible unmeasured confounder with respect to the E-values of Table S2. If there is no space in the main paper, it would be nice to have some expanded discussion of the interpretation of the E-values in the supplement.

2. I feel the conclusions about no differences between the two vaccines with respect to disease and death were vague and overstated in both the abstract and the conclusion of the paper. For example, the abstract states that there "were no differences in vaccine efficacy for protection against hospitalization, ICU admission, or death/hospice transfer". The results section is more clear, stating that "while events are rare, no statistically significant differences between the two vaccines were observed". I think the rarity of the events is very important for the interpretation of the non-significance. Non-significance typically means that the confidence intervals do not contain the null effect (e.g., the adjusted odds ratio value of 1), but among confidence intervals that contain the null effect, there are confidence intervals that are either very wide (meaning the study did not have enough information to say much about that effect) or very narrow (meaning that the study was well powered to see the effect, but saw none), or anywhere in between. The reader needs to go to the supplement to see how precise the effects were estimated. It would be better to pick a few of the most appropriate results from Figure S3, and give the aOR estimates and 95% CI. For example, the last row of Fig S3 gives for Hospitalization/ICU/Death/Hospice and adjusted odds ratio estimate of 0.91 (95% CI: 0.53, 1.59). A statement like that in the abstract would be more precise and useful, than just saying there were "no differences in vaccine efficacy".

Minor Comments

1. Could you please describe in more detail the truncating statements in Figure 1 in the text. For example, does "truncating on May 1, 2021" mean that only individuals fully vaccinated (i.e., 14 days after second dose) on or before May 1, 2021 are included in the database? I think this is what it says in lines 216-221 of the Supplement. I am just a little unsure because the term "truncation" has a specific meaning in the time-to-event statistical literature. In that literature, truncation means that you will not observe some individuals' time to event if they do not fall into a certain time frame (usually measured in time scale of interest, which in your case is time since fully vaccinated), and you do not know how many individuals you are missing. But your use of the term appears to be slightly different.

2. In Figure 3, Figure S8, S9, S11, and especially in Figure S3, S4, and S5, the marginal probabilities are given. It would be better to give those marginal probabilities multiplied by 100,000 or 1,000,000, so that you can better differentiate between the Moderna and Pfizer marginal probabilities (e.g., list the probabilities analogously to rates per 100,000).

3. In Figure 1, the orange box: I do not understand the phrase "Excluding patients experiencing Covid related events....within 13 days of second dose but no events after 14th day". Why don't you exclude all patients that have Covid related events within 13 days second dose? Why do you need to require additionally that they had no events after 14th day?

4. You mention the Delta VOC in both the introduction and the discussion. I found myself wondering

how much the Delta variant was prevalent in the US during the time of your study, given your data end by May 31, 2021. Was the Delta VOC an important factor during the time frame of your study? Perhaps you could give some context about that in the discussion. Or perhaps all of that is a digression from your main point. Similarly, at the end of your introduction you mention the important problem of when booster dose are recommended. Does your analysis directly help with that problem? If not, perhaps do not mention it.

5. In the Supplemental table on line 248, the "Adjustment", "IPTW" and "Subclassification" columns are unreadable in my version of the paper.

We are grateful to the editor and to the two reviewers for their thorough reading of our paper and insightful comments. We hope that upon revision, the editor will find this manuscript acceptable for publication.

A major change in this revision was an update to our data which meant a large increase in our population of vaccinated individuals. This is due to the way in which we receive vaccination information which, when delivered by states, is often done on an irregular (quarterly, monthly, etc) basis. In addition to new individuals in the population, our data is updated as new claims are submitted and adjudicated. Finally, in response to the reviewer's excellent suggestion to include information on individuals' location (urban, suburban, or rural) we removed individuals from our population for whom this data was not available. We have kept our study period the same so as to not interact with the introduction of the delta and omicron variants in the US.

Reviewer #1 (Remarks to the Author):

General Comments:

This paper presents useful relative effectiveness results for the two key mRNA vaccines used in the USA, Moderna and Pfizer, with respect to infection in those with health-seeking behavior and a composite serious disease index. The large data set with extensively documented demographic information and comorbidities along with vaccination date enable this rigorous analysis, that uses advanced causal inference tools to adjust for potential confounding.

The results have potential import in providing relative effectiveness information, although the ability of the results to inform prioritization and decision making for boosters is limited by the fact that results did not compare with unvaccinated controls or compare early/late periods to assess decrease magnitude of potential immune protection. Further, the study does not adjust for infection time effects, especially as related to the emergence of Delta in the summer. This could potentially confound the inter-vaccine comparison, and clearly Delta is confounded with time since vaccination, which makes the results more difficult to interpret. This should at a minimum be discussed more thoroughly, and preferably adjusted for in the analysis. Also, a sub analysis including days 0-13 and another for previously infected subset, although underpowered, would provide useful point estimates that would be relevant and useful.

Delta variant prevalence as of May 31, 2021 (we censored at June 1, 2021) was less than 7% of cases in the US (<https://ourworldindata.org/grapher/covid-variants-area?country=~USA>). So, although we agree with the reviewers that infections caused by the delta variant are of importance, we do not feel this is relevant to the data we've presented.

We agree that sub-analyses of days 0-13 after second vaccine and for those who have evidence of a prior COVID-19 infection would be of interest but, because the event rate in the necessarily smaller samples would be so low, we do not believe any statistically or clinically relevant information will be gained.

Specific comments:

- *The details of the IPTW model should be presented in the supplement.*

Supplement 3: Methods section B1 is on the IPTW method. This begins on page 21. We are happy to provide further information if the editor requests.

- *What are the characteristics of the Medicare Advantage and Commercially Insured cohorts? They are combined together, but are fundamentally different in many ways. The authors should:*
 - o *Put together a table of demographic and clinical factors for the two cohorts to show their differences.*

We have recreated Table 1 for those who are less than 65 and 65 or over separately. See Tables S3 and S4. While this is not an exact match to the reviewer's request to look at Medicare and Commercial enrollees separately, it does match with their request below to repeat our analysis in the over 65 group with the under 65 group. We hope this modification will answer the reviewer's question. In addition, we added rows detailing insurance type within Table 1 and in the relevant newly requested tables.

- o *Perform a secondary analysis on each of these cohorts separately to see whether the vaccine differences, and the vaccine by time interaction, differ across these cohorts.*

While we did not do this by Medicare versus Commercial, we did look separately at those aged 65+. See what is now Figure S9. In addition, we are adding a Figure to show results in the cohort of those under 65 (Figure S10).

- *Minor point: Table 1 should split out age by 5-year groups, not just mean and variance, to show the distribution of this key factor across vaccine groups. Do the same for the supplemental tables split by Medicare/commercial cohorts.*

Thank you for this suggestion which we have implemented.

- *Show p-values for comparisons in Table 1 between the vaccinated group for various factors (could be a supplemental table if it doesn't fit in the main table)*

Since these would be unadjusted p-values we do not feel they provide significant insight and would prefer not to include this additional analysis.

- *The presented results focus on the NNV, which is useful but whose scale depends heavily on the context and is difficult to compare with results from other studies looking at relative rates. The text and abstract should also present the HR and 95% CI for each comparison. For the non-statistically-significant composite index, still present the estimated HR and 95% CI to show the estimated effect size and precision. I know that the supplement contains results for many different model choices/adjustments – pick whichever one is considered the primary one and make that the one whose HR and 95% intervals you present.*

At the reviewers useful suggestion, we have added this information to the discussion section.

- *It is helpful that the secondary analysis of “not previously infected” was done, but the authors should also add a secondary analysis including only the “previously infected” cohort. This will be smaller sample size and underpowered but given the lack of data on this endpoint in many existing studies, even a point estimate and 95% CI could help demonstrate vaccine differences in previously infected.*

As the reviewer points out, the group that is “previously infected” would give an underpowered analysis. In addition, the data would be skewed toward those whose

infection was serious since we are missing a portion of testing data (including any tests done at home) but would see if people were hospitalized. We, unfortunately, feel this is outside the scope of our work at this time.

- *Similarly for the age stratification – a separate secondary analysis was done in the >65 in addition to the overall. A ≤ 65 cohort analysis should also be added to the secondary analysis section in the supplementary materials.*

This is a useful suggestion which we have added to our supplement as noted above. See table S4 and Figure S10.

- *The modeling started at 14 days after 2nd dose, which is standard since that is when the “full protection” kicks in. There is a paucity of data on advanced endpoints like hospitalization, ICU, death in your composite endpoint from day 0 of dose 1 until day 14 to dose 2, and certain people in society and social media try to claim this is done to conceal evidence of dangerous vaccine side effects. Please also include a secondary analysis of your infection and especially composite serious disease endpoint during this time frame – even if underpowered, the point estimates and transparency of showing it will be a contribution to the literature, in my opinion.*

We agree with the reviewer that information on vaccine side effects would be a valuable contribution to the literature. Unfortunately, this is outside the scope of our study.

- *The model rigorously adjusts for many of the key confounders, which is a significant strength of the paper. However, there is no adjustment for time of infection, a key confounder especially given the emergence of Delta in summer 2021. Is there some way to adjust for time confounding in the modeling?*

As stated above, given the censoring of our data to June 1, 2021, we do not think the delta variant comes into play here. Time to infection is also one of the responses of interest and the survival regression based on time-to-event analysis explicitly uses the time of infection as an event.

- *Related to this concern about time effects, I have the following questions/suggesting/requests:*
 - o *What is the time frame of follow up period, and in particular does it include Delta periods and pre-Delta periods?*

The follow up time periods are shown in our CONSORT diagram on page 14. We had trustworthy data through June 1, 2021 to observe our outcomes. So, for the 30-day response, we truncate on May 1, 2021, for the 60-day response we truncate on April 1, 2021 and the 90 day response we truncate on March 2, 2021.

- o *Are potential calendar time effects modeled? It doesn't look like it.*

Calendar time is considered. We adjust for “days to vaccination” in all our modeling. See Table S1.

- o *Is the distribution of vaccination times over time for the two vaccines consistent? Plot a histogram of vaccinations over time for the two to see if they match up over time. If very different, this is a potential confounder and should be discussed.*

We thank the reviewer for this useful question. To answer it we have added Figure S2 which shows calendar date on the x-axis and doses of each vaccine over time.

- *While the comparison between the two vaccines is useful, and seeing the difference increases over time is especially interesting, the lack of formal comparison across time regions since vaccination and the lack of unvaccinated controls makes it more difficult to interpret.*

- o *Ideally, it would be great to pull out a matched unvaccinated cohort against which to compare each of the vaccines and time since vaccination to give an idea of vaccine effectiveness and its waning over time. It is not clear whether after waning the Pfizer loses most all of its effectiveness or not. I recognize this might be a lot of work, but would add substantially to the paper if done.*

- We would love to look at an unvaccinated control group. Unfortunately, the lack of presences of a vaccination in our data does NOT imply that a person did not receive a vaccine. For instance, we are missing most vaccination that occurred at mass-vaccination sites. Hence, we cannot create a reliable control group of unvaccinated individuals.

- o *The days 30-60 and 60-90 infection rates should be compared with the days 0-30 after 14 days post vaccination infection rates to quantify the increased hazard ratio evident in the waning/Delta effect.*

- We agree this is an interesting comparison and feel that we comment on this phenomena by noting that the difference in performance increases over time. In Fig 3 for instance, the NNV and marginal probabilities of experiencing events at 10 days, 30 days, 50 days, 70 days, and 90 days are all listed next to each other.

- *By the way, the comparison of days 60-90 to days 0-30 could be due to waning, Delta, or some combination. This should be thoroughly discussed, and if possible, proportion of delta in the population used as a covariate in the analysis.*

- Given delta's low prevalence in the US at the time of our data collection, we believe this is due to waning immunity provided from the vaccine.

- *Discussion line 249 “potential risks for booster doses, particularly if immune related side effects more frequent than prevented serious infections” – transient side effects not equivalent to hospitalized side effects – reword and if suggesting vaccines increase probability of serious adverse events, provide citation – the clinical trials did not find this relative to placebo in either study.*

- We thank the reviewer for this suggestion. Since boosters have now been recommended by the CDC for everyone 18 and over for all three vaccines, we find this paragraph in our paper is no longer relevant and have deleted it.

Reviewer #2 (Remarks to the Author):

This paper compares differential vaccine efficacy of between the mRNA-1273 (Moderna) and the BNT162b (Pfizer) COVID-19 vaccines. The analysis is based on 3.5 million patient records from a single US insurer. The analyses controlled for many factors using appropriate statistical methods (e.g., using multivariable logistic regression with inverse probability of treatment weights based on propensity scores). The analyses were very thorough, and included sensitivity analyses to explore the assumptions. The paper addresses an important issue, and the analyses help shed some light on it. I have a few comments.

Major Comments

1. I feel the study limitations paragraph in the discussion section could be expanded. My main concern is that there may have been some important differences between the vaccine groups that were not measured (i.e., there may have been some unmeasured confounders). One possible unmeasured confounder is degree of urban/rural of residence. One major difference between the vaccines is the necessary refrigeration. The BNT162b vaccine requires more intense refrigeration, and some rural hospitals could not afford that (see e.g., <https://www.statnews.com/2020/11/11/rural-hospitals-cant-afford-freezers-to-store-pfizer-covid19-vaccine/>). So there may be an urban/rural difference in the distribution of the two vaccines. Although the authors mentioned adjusting for residence by state, within each state there are both urban and rural areas. Additionally, there may have been a substantial urban/rural difference in SARS-CoV-2 infection rates during the time the study was covering. Perhaps you could discuss this possible unmeasured confounder with respect to the E-values of Table S2. If there is no space in the main paper, it would be nice to have some expanded discussion of the interpretation of the E-values in the supplement.

We appreciate the reviewer's comment on rural/ urban and how it may relate to vaccine distribution. We have added this information on each individual and re-run our models to include this feature.

2. I feel the conclusions about no differences between the two vaccines with respect to disease and death were vague and overstated in both the abstract and the conclusion of the paper. For example, the abstract states that there "were no differences in vaccine efficacy for protection against hospitalization, ICU admission, or death/hospice transfer". The results section is more clear, stating that "while events are rare, no statistically significant differences between the two vaccines were observed ". I think the rarity of the events is very important for the interpretation of the non-significance. Non-significance typically means that the confidence intervals do not contain the null effect (e.g., the adjusted odds ratio value of 1), but among confidence intervals that contain the null effect, there are confidence intervals that are either very wide (meaning the study did not have enough information to say much about that effect) or very narrow (meaning that the study was well powered to see the effect, but saw none), or anywhere in between. The reader needs to go to the supplement to see how precise the effects were estimated. It would be better to pick a few of the most appropriate results from Figure S3, and give the aOR estimates and 95% CI. For example, the last row of Fig S3 gives for Hospitalization/ ICU/ Death/ Hospice and adjusted odds ratio estimate of 0.91 (95% CI: 0.53, 1.59). A statement like that in the abstract would be more precise and useful, than just saying there were "no differences in vaccine efficacy".

We appreciate the reviewer's request for further specificity and we have added the aOR and confidence intervals to the abstract and conclusion as suggested.

Minor Comments

1. *Could you please describe in more detail the truncating statements in Figure 1 in the text. For example, does "truncating on May 1, 2021" mean that only individuals fully vaccinated (i.e., 14 days after second dose) on or before May 1, 2021 are included in the database? I think this is what it says in lines 216-221 of the Supplement. I am just a little unsure because the term "truncation" has a specific meaning in the time-to-event statistical literature. In that literature, truncation means that you will not observe some individuals' time to event if they do not fall into a certain time frame (usually measured in time scale of interest, which in your case is time since fully vaccinated), and you do not know how many individuals you are missing. But your use of the term appears to be slightly different.*

Thank you for pointing this out. We've changed our wording in the figure to "fully vaccinated by xxx" which, we hope, is more clear. We've also made adjustments in captions.

2. *In Figure 3, Figure S8, S9, S11, and especially in Figure S3, S4, and S5, the marginal probabilities are given. It would be better to give those marginal probabilities multiplied by 100,000 or 1,000,000, so that you can better differentiate between the Moderna and Pfizer marginal probabilities (e.g., list the probabilities analogously to rates per 100,000).*

Thank you for this useful suggestion. We've changed the marginal probabilities to predicted marginal rates per 1,000 people which we hope will make the results more easily interpretable.

3. *In Figure 1, the orange box: I do not understand the phrase "Excluding patients experiencing Covid related events....within 13 days of second dose but no events after 14th day". Why don't you exclude all patients that have Covid related events within 13 days second dose? Why do you need to require additionally that they had no events after 14th day?*

We wanted to include people who have a Covid-related event after they are fully vaccinated (14 days after dose two) regardless of whether they had a Covid-related event immediately following their vaccination but before they'd achieved "fully vaccinated" status. However, if someone only experienced an event in days [0, 13] relative to dose 2, we cannot be sure if this is related to a side effect or contracting the disease and so we excluded these individuals. In addition, we did a sensitivity analysis where we removed anyone with any diagnosis of COVID-19 prior to being "fully vaccinated" and results are presented in Figure S11.

4. *You mention the Delta VOC in both the introduction and the discussion. I found myself wondering how much the Delta variant was prevalent in the US during the time of your study, given your data end by May 31, 2021. Was the Delta VOC an important factor during the time frame of your study? Perhaps you could give some context about that in the discussion. Or perhaps all of that is a digression from your main point. Similarly, at the end of your introduction you mention the important problem of when booster dose are recommended. Does your analysis directly help with that problem? If not, perhaps do not mention it.*

We thank the reviewer for their suggestion which we have taken. We deleted the mention of delta in the introduction and generalized to all variants of concern. We also removed the paragraph regarding booster doses and replaced it with a paragraph noting delta's low prevalence during the period of our study and suggesting that further comparative analyses should continue as new variants emerge.

5. In the Supplemental table on line 248, the "Adjustment", "IPTW" and "Subclassification" columns are unreadable in my version of the paper.

Thank you for pointing this out. We hope this will be fixed in publication.

Reviewers' Comments:

Reviewer #1:

Remarks to the Author:

We thank the authors for their response and making changes in response to some of the concerns and suggestions.

Unvaccinated controls and previously infected, and days 0-13: While the lack of these limits what can be learned from this paper, the difficulties and limitations of the data are understood.

Results: thank you for adding HR and CI to the discussion, but to help the reader quickly grasp the results, the HR and CI for key results should be in the results section, not just the discussion.

Time confounding: Thank you for clarifying that the study ended before the Delta surge. BTW this should be mentioned in the Discussion to highlight that this paper addresses vaccine differences during the Alpha and pre-Alpha eras. I appreciate that the model accounts for time since vaccination, and includes vaccination time (+/- Feb 1), and that Delta is not a confounder, but I am still not sure that calendar time confounding is sufficiently accounted for here. Even without Delta, over the time period of the study, the variant distribution shifted substantially from pre-alpha to Epsilon/Alpha up to the time when Delta took over. It seems this could confound some of the comparisons, especially related to later times since vaccination for which a higher proportion of the infections would be of these more transmissible variants. I still think more needs to be done to either demonstrate this is not an issue, adjust for it, or at least provide some summaries of time of infection in the 3 time groups x 2 vaccine groups and discuss the potential limitation.

Illustration of waning effect over time: While Figure contains information at the difficult time points, it is very difficult to read -- so much information with all endpoints and analyses, and summarizing aHR, NNV, and predicted marginal rates. It is not clear to me that all of the different analyses should be in the figure in the main paper -- it would be much better to include just the unadjusted and whatever you consider the "best" adjusted analysis in the main figure and put the others in the supplement. Also, some graphical summary to make it easy to follow the waning effect as well as the differences between vaccines predicted by the model is necessary for the reader to appreciate the results and quickly see the key ones -- perhaps the predicted marginal rates per 1k should be what is plotted. Also, as said above, the aHR and CI for the key adjusted model for each of the 3 outcomes should be presented in the results (and preferably abstract too)

Reviewer #2:

Remarks to the Author:

The authors have addressed most of my concerns. I appreciate the work that went into redoing the analysis with the rural/urban adjustment added. I have a few minor comments.

1. In the abstract, you gave an adjusted OR of "(aOR 1.23, 95% CI (0.67, 2.25))". When I looked up that value in Figure S6, the row that gives that odds ratio is labeled "Unadjusted and with IPTW". Since the value is an unadjusted odds ratio, you should put "OR" instead of "aOR". More to the point, why is an unadjusted odds ratio instead of an adjusted OR the most important result that is presented in the abstract? Isn't it better to adjust, given that this is an observational study not a randomized study?

2. Typo, Figure 1, orange box: "mRNA-1273=1,59,450". Missing digit, "1,59?,450" ?

3. Typo, Figs S5 and S6: there are two "model" columns, with the second one overlaid over the first.

We are grateful to the editor and to the two reviewers for their second thorough reading of our paper and insightful comments. We have addressed all the comments and hope that upon revision, the editor will find this version of the manuscript acceptable for publication.

Reviewer #1 (Remarks to the Author):

We thank the authors for their response and making changes in response to some of the concerns and suggestions.

Unvaccinated controls and previously infected, and days 0-13: While the lack of these limits what can be learned from this paper, the difficulties and limitations of the data are understood.

Results: thank you for adding HR and CI to the discussion, but to help the reader quickly grasp the results, the HR and CI for key results should be in the results section, not just the discussion.

Thank you for this suggestion. We have added this information in the fourth paragraph of the results section.

Time confounding: Thank you for clarifying that the study ended before the Delta surge. BTW this should be mentioned in the Discussion to highlight that this paper addresses vaccine differences during the Alpha and pre-Alpha eras. I appreciate that the model accounts for time since vaccination, and includes vaccination time (+/- Feb 1). and that Delta is not a confounder, but I am still not sure that calendar time confounding is sufficiently accounted for here. Even without Delta, over the time period of the study, the variant distribution shifted substantially from pre-alpha to Epsilon/Alpha up to the time when Delta took over. It seems this could confound some of the comparisons, especially related to later times since vaccination for which a higher proportion of the infections would be of these more transmissible variants. I still think more needs to be done to either demonstrate this is not an issue, adjust for it, or at least provide some summaries of time of infection in the 3 time groups x 2 vaccine groups and discuss the potential limitation.

We're grateful to the reviewer for this question. In our paper, we focus on "time" as days since full vaccination. In addition, calendar time, which relates to the likely variant with which a patient is infected, is also of importance. Our analyses do include an attempt to understand pre-alpha and alpha variant dominant effects via a binary calendar time (e.g., before or after Feb 1). This assumption, while simple and interpretable, is likely too restrictive to capture everything since the relationship between vaccines and variants is complex and non-linear. A more sophisticated model with smooth time-varying effects addressing the interaction between vaccines and variants is warranted. While this is beyond the scope of this paper, we leave this for our future research.

In order to address the reviewer's concern, we have added comments about this limitation in the discussion section. We include the table below in the supplementary materials which relates the incidence of each of the study outcomes over the early, pre-alpha, and alpha-dominant phases of the pandemic.

Table S7: Summary tables for responses by time-period which roughly correspond to early-, pre-alpha, and alpha-dominant periods of the pandemic.

Study-related outcomes	Before February 28	March 1- March 31, 2021	April 1-May 1, 2021
---------------------------	--------------------------------	----------------------------

	BNT162B2	MRNA-1273	BNT162B2	MRNA-1273	BNT162B2	MRNA-1273
Fully Vaccinated	199069	149466	361072	415817	820430	493420
Infection; N (%)	1659 (0.8)	454 (0.3)	901 (0.2)	753 (0.2)	1206 (0.1)	742 (0.2)
Hospitalization/ICU/ deceased/ transferred to hospice; N (%)	35 (0.0)	7 (0.0)	15 (0.0)	19 (0.0)	22 (0.0)	16 (0.0)
ICU/deceased/ transferred to hospice; N (%)	17 (0.0)	1 (0.0)	5 (0.0)	8 (0.0)	10 (0.0)	5 (0.0)
Hospitalization (%)	30 (0.0)	6 (0.0)	11 (0.0)	16 (0.0)	17 (0.0)	13 (0.0)
ICU; N (%)	14 (0.0)	1 (0.0)	5 (0.0)	8 (0.0)	9 (0.0)	5 (0.0)
Deceased/ transferred to hospice; N (%)	7 (0.0)	0 (0.0)	0 (0.0)	1 (0.0)	2 (0.0)	1 (0.0)

Illustration of waning effect over time: While Figure contains information at the difficult time points, it is very difficult to read -- so much information with all endpoints and analyses, and summarizing aHR, NNV, and predicted marginal rates. It is not clear to me that all of the different analyses should be in the figure in the main paper -- it would be much better to include just the unadjusted and whatever you consider the "best" adjusted analysis in the main figure and put the others in the supplement. Also, some graphical summary to make it easy to follow the waning effect as well as the differences between vaccines predicted by the model is necessary for the reader to appreciate the results and quickly see the key ones -- perhaps the predicted marginal rates per 1k should be what is plotted. Also, as said above, the aHR and CI for the key adjusted model for each of the 3 outcomes should be presented in the results (and preferably abstract too)

We thank the reviewer for this excellent suggestion. We have trimmed Figure 3 to include just the two best models. In addition, we added panel b which plots the number needed to vaccinate over time and clearly shows that the difference between the two vaccines grows (number needed to vaccinate decreases) as days since full vaccination increase. Finally, we added aHR and 95% CI to the fourth paragraph of the results section.

Reviewer #2 (Remarks to the Author):

The authors have addressed most of my concerns. I appreciate the work that went into redoing the analysis with the rural/urban adjustment added. I have a few minor comments.

1. In the abstract, you gave an adjusted OR of “(aOR 1.23, 95% CI (0.67, 2.25))”. When I looked up that value in Figure S6, the row that gives that odds ratio is labeled “Unadjusted and with IPTW”. Since the value is an unadjusted odds ratio, you should put “OR” instead of “aOR”. More to the point, why is an unadjusted odds ratio instead of an adjusted OR the most important result that is presented in the abstract? Isn't it better to adjust, given that this is an observational study not a randomized study?

This corresponds to the weighted model with IPTW which is calculated using propensity scores adjusted for all observed confounders. The corresponding weights in the outcome model (which is though an unadjusted outcome model) aim to correct the imbalances between treatments with respect to observed covariates and thus tries to accomplish the same goal of causal inference as randomized experiments do. We want to believe that the underlying vaccine effect should be same irrespective of patient characteristics – and thus the reported OR from the weighted model may be a more representative estimate for the inherent real-world evidence. Therefore, for consistency and simplicity, we reported this particular OR. Note weighting may not always bring optimal results and may indeed (de)increase the bias in estimated causal parameters. To validate our inferential results, we fitted different variations of models with(out) IPTW. Irrespective of different model specifications, we obtain similar inference for vaccine effects signaling the consistency of results.

2. Typo, Figure 1, orange box: “mRNA-1273=1,59,450”. Missing digit, “1,59?,450” ?

Thank you for pointing this out. We have fixed our error, with the corrected label of sample size reading mRNA-1273 = 1,598,450.

3. Typo, Figs S5 and S6: there are two “model” columns, with the second one overlaid over the first.

Thank you, we have corrected this mistake.